# Micro-Computed Tomographic Assessment of Microcrack Formation before and after Instrumentation of Curved Root Canals with Neoniti Rotary Files

**DOI:** 10.3390/ma15093002

**Published:** 2022-04-20

**Authors:** Parichehr Zarean, Mutlu Özcan, Paridokht Zarean, Seyed Omid Haghani, Maryam Zare Jahromi, Nadin Al-Haj Husain, Masoud Khabiri

**Affiliations:** 1Department of Oral and Cranio-Maxillofacial Surgery, University Hospital Basel, 4031 Basel, Switzerland; pch.zarean@gmail.com (P.Z.); p.zarean@gmail.com (P.Z.); 2Division of Dental Biomaterials, Clinic for Reconstructive Dentistry, Center of Dental Medicine, University of Zurich, 8032 Zurich, Switzerland; mutluozcan@hotmail.com (M.Ö.); nadin.al-haj-husain@zmk.unibe.ch (N.A.-H.H.); 3Department of Endodontics, School of Dentistry, Isfahan (Khorasgan) Branch, Islamic Azad University, Isfahan 81551-39998, Iran; dr.omid.h.1372@gmail.com (S.O.H.); m.khabiri@khuisf.ac.ir (M.K.); 4Department of Reconstructive Dentistry and Gerodontology, School of Dental Medicine, University of Bern, 3010 Bern, Switzerland

**Keywords:** curved root, micro-computed tomography, microcrack, Neoniti, root canal therapy

## Abstract

The aim of this study was to assess the microcrack formation of moderately and severely curved root canals following instrumentation with Neoniti rotary files using micro-computed tomography. This in vitro study evaluated 18 extracted sound mandibular molars with two separate mesial canals and foramina in two groups (n = 9) with 5–20° (moderate) and 20–40° (severe) root canal curvature. The number of microcracks in the root canal walls was counted at baseline by micro-CT. Subsequently, the root canals were instrumented with 0.20/0.06 v Neoniti files as single files with a torque of 1.5 Ncm and a speed of 400 rpm. The number of microcracks was counted again postoperatively on micro-CT images using Amira software. Statistical analysis was performed using the Shapiro–Wilk test, Levene’s test and repeated-measures ANOVA (α = 0.05). The mean number of microcracks significantly increased postoperatively in both the moderately curved (11.59 ± 9.74 vs. 8.2 ± 7.4; *p* = 0.001) and the severely curved (13.23 ± 5.64 vs. 7.20 ± 5.94; *p* < 0.001) groups. However, the differences between the two groups were not significant (*p* = 0.668). Based on the results obtained, it can be stated that the instrumentation of moderately and severely curved root canals with Neoniti rotary files increases the number of microcracks. However, the higher degree of curvature does not necessarily translate to a higher number of microcracks after root canal instrumentation with this specific rotary system and methodological procedures.

## 1. Introduction

In endodontic treatment, the anatomical complexity of curved root canals makes it challenging to produce instrumentation that is both effective and free of iatrogenic events [1]. Currently, one of the significant causes of tooth loss is root fracture, which has become a major concern in endodontics [2,3].

There are several theoretical etiologies for root fractures. Some hypotheses state that root fractures initiate from the dentinal microcracks caused by excessive forces during filling procedures, post placement, the designs of spreaders and the dehydration of dentin [2,4,5,6]. In addition, constant occlusal forces may lead to microcracks, craze lines, or vertical root fractures [7,8].

Some studies demonstrated that there might be a relationship between microcrack formation and the process of canal preparation with rotary instruments [9]. Adorno et al. also reported significant root weakness upon root canal preparation, which might also be the reason for apical cracks [10].

In the past two decades, different rotary and reciprocal systems with unique designs have been introduced to the market, aiming to speed up and simplify the process and improve the quality of root canal treatment. The introduction of nickel titanium (NiTi) rotary files in 1988 and the subsequent development of rotary NiTi files contributed to the optimal preparation of curved root canals. Despite the several advantages of NiTi rotary files (e.g., lower risk of procedural errors, including ledge formation, zipping, canal transportation and canal perforation due to higher flexibility and the specific geometric design of each file) [11,12,13], evidence shows that NiTi rotary files might damage dentin and cause cracks in root canal walls [14,15,16].

The third generation of NiTi rotary files are manufactured by using heating and cooling procedures on wires and by applying M-wire and R-phase technologies and electrical discharge methods, which results in a reduction in the separation risks of the files, especially in curved root canals, as well as high memory shape, a reduction in cyclic fatigue and the greater preservation of the initial anatomy of the root canal. Neoniti (Neolix, Châtres-la-Forêt, France) is one of the instruments that has been introduced as part of this generation. It features full rotary motion and multiple tapers in a single instrument, which consists of one C1 for coronal enlargement and three A1 for shaping canal (with 0.2, 0.25 and 0.4 tip sizes) files [17,18,19,20].

Among several methods that have been proposed for the assessment of root canal morphology and the quality of root canal treatment, such as serial sectioning, electron microscopy, radiographic comparison, microcomputed tomography (micro-CT) and cone-beam computed tomography, micro-CT is a non-invasive technique for the three-dimensional assessment of root canal morphology in vitro. This technique is highly accurate and provides high-resolution images that enable precise assessments. The method does not destroy samples; thus, it enables the comparison of samples before and after interventions [9,21,22,23,24,25,26,27,28].

To the best of the authors’ knowledge, studies regarding the effect of canal curvature on the formation of microcracks following the use of different rotary systems are limited. Considering the difficult preparation of narrow curved canals, the significance of the speed and quality of root canal instrumentation and the prevention of procedural errors, as well as the increasing use of Neoniti files for the preparation of curved canals, this study aimed to assess microcrack formation in moderately and severely curved root canals before and after instrumentation with Neoniti rotary files using micro-CT. The null hypothesis is that there were no significant differences between the curvature of the root canals and the creation of micro-cracks before and after instrumentation using Neoniti rotary files.

## 2. Materials and Methods

This in vitro study evaluated mandibular molars that were freshly extracted with two separate mesial canals and foramina for several reasons. Extracted mandibular molars with cracks, resorption or fracture, open apices, calcification of the mesiobuccal canal, internal root resorption and history of previous endodontic treatment were excluded from the study. The study protocol was approved by the ethics committee and institutional review board of the university (IR.IAU.KHUISF.REC.1398.210). After collection, the teeth were disinfected with 5.2% sodium hypochlorite and were inspected under a ×10 magnification stereomicroscope equipped with a digital camera (SMP 200, HP, CA, USA) from the mesial, buccal and lingual aspects to ensure absence of cracks in the mesiobuccal roots. The teeth were also radiographed and the mesiobuccal canal curvature was measured using Schneider’s method [29]. Molar teeth with mesiobuccal root curvature between 5–20° and 20–40° were selected by convenience sampling and stored in saline at 4 °C until further examination.

For measurement of the root canal curvature according to Schneider’s method, the teeth were fixed on wax and underwent digital radiography (Kavo, Santa Catarina, Brazil) mesiodistally at 2-millimeter distance by the parallel technique. All radiographs were then analyzed by Protractor Android software to measure the degree of root curvature (Figure 1 and Figure 2). The sample size was calculated to be nine teeth in each group (α = 0.05). Teeth with 5–20° root canal curvature were assigned to the moderately curved group (n = 9) while those with 20–40° root canal curvature were assigned to the severely curved group (n = 9).

An access cavity was prepared using a #4 conical fissure bur (Dentsply, Maillefer Ballaigues, Switzerland) and the mesiobuccal canal orifice was negotiated using a #10 K-file (Mani, Japan). The teeth that did not allow access of the #10 K-file to the apical foramen were excluded and replaced. The teeth in each of the two groups were inversely mounted in a Teflon mold to the level of their cementoenamel junction (CEJ). In order to scan the teeth with maximum field of view and eliminate the superimposition of images, Teflon molds with 7 cm diameter and 3 cm height with three holes were prepared. The teeth were randomly mounted in the holes parallel to each other and parallel to the mold walls using auto-polymerizing acrylic resin (Pyrax, India). All molds containing teeth underwent micro-CT (Sky Scan, Belgium) with 35-micrometer pixels, at a voltage of 80 kV, for 25 min. Baseline scanning was performed to find primary microcracks in the root canals prior to instrumentation. Before the chemomechanical instrumentation and after determining the working length visually, roots were coated by hydrophilic vinyl polysiloxane impression material (Express XT; Neuss, Germany) and subsequently embedded in auto-polymerizing acrylic resin to simulate the periodontal ligament [30]. Next, one operator instrumented the root canals in both groups as follows. The pulp chamber was filled with 5.25% sodium hypochlorite (NaOCl) irrigating solution and a #15 K-file (Mani, Japan) with 0.02 taper was introduced into the canal to the working length. Subsequently, C1 Neoniti file (Neolix; Châtres-la-Forêt, France) with torque of 1.5 Ncm and speed of 400 rpm was used along with a Meta micromotor (Meta, South Korea) to shape the coronal third of the canal to create a straight path for the next file. Next, 0.2 Neoniti file (Neolix; Châtres-la-Forêt, France) was used with the Meta micromotor with settings similar to those of an orifice shaper. It was used as a single file to the working length. After each filing, the canal was rinsed with 5.25% sodium hypochlorite and saline. Each file was used three times and then discarded. Finally, to ensure root canal preparation with the desired taper, a #25 gutta-percha with 4% taper was placed in the canal. If it reached the working length, the root canal preparation was sufficient.

After root canal preparation, the teeth underwent micro-CT evaluation again and the images were then processed by Amira software (Berlin, Germany). The number of microcracks in the two groups was measured on images (Figure 3 and Figure 4) taken before and after root canal instrumentation [31]. The microcracks detected on preoperative and postoperative images of each tooth were counted. The number of images (slices) of each tooth showing microcracks was counted and divided by the total number of images of the respective tooth to determine the ratio of images showing microcracks. Due to small differences in the root lengths of the teeth, small differences were present in the number of images of each tooth, which was taken into account in statistical analysis. It should be noted that all images were evaluated by Amira software, which is a powerful, multifaceted 2D–5D solution for visualization, processing and analysis of research data from many image modalities. The evaluation was performed by of two qualified endodontists with about 30 years’ experience in the field of endodontology.

Repeated-measures ANOVA was applied for simultaneous assessment of the effects of instrumentation and degree of curvature on the number of microcracks. The Shapiro–Wilk test was used to analyze normal distribution of data and Levene’s test was applied to assess the homogeneity of variances. To assess the homogeneity of the variance matrix for ANOVA, the box test was applied. The Bonferroni post hoc test was utilized to compare the number of microcracks before and after instrumentation in each group, as well as between the two groups before and after instrumentation. All statistical analyses were carried out using SPSS version 24 at 0.05 level of significance.

## 3. Results

The measurements of the central dispersion of the mean number of microcracks in the root canal walls of the moderately curved canals before and after instrumentation are shown in Table 1. Table 2 presents the equivalent measurements for the severely curved canals. The repeated-measures ANOVA revealed that the effect of instrumentation on the number of microcracks was significant. However, the effect of the degree of curvature (*p* = 0.945) and the effect of the interaction between the instrumentation and the degree of curvature (*p* = 0.671) on the number of microcracks were not significant. The results of the Bonferroni post hoc test also showed that the mean number of microcracks significantly increased postoperatively in both the moderately curved (*p* = 0.001) and the severely curved (*p* < 0.001) groups. On average, the ratio of images with microcracks increased by 4.2% after instrumentation in the moderately curved group, while this value was 6.0% in the severely curved canals. However, the mean number of microcracks was not significantly different between the two groups before (*p* = 0.955) or after (*p* = 0.668) instrumentation.

## 4. Discussion

This study aimed to assess the microcrack formation in moderately and severely curved root canals following instrumentation with Neoniti rotary files using micro-CT. The mesiobuccal canals of mandibular molars were evaluated since they are among the most difficult root canals in which to perform endodontic treatment due to their constricted anatomical configuration, inherent concavities, complexities and curvature. They have a greater potential to develop cracks and higher rates of fracture during mechanical preparation as a result of the root-canal dentinal-wall thinness in some of their parts and their smaller dimensions [27,30,32,33,34,35].

The results showed a significant increase in the number of microcracks in both groups after instrumentation, compared with baseline. However, the difference in this respect was not significant between the two groups, of moderately curved and severely curved canals. Thus, the null hypothesis was partially rejected.

These results are in agreement with a study by Khoshbin et al. [35], in which an increase in the number of microcracks in all rotary groups (Neoniti, ProTaper, Reciproc and Mtwo) compared with the control group was reported. Tomer et al. [36] also compared the Neoniti, ProTaper Universal and 2Shape rotary systems and found that all rotary files caused microcracks, but that Neoniti files created fewer microcracks in comparison to the other rotary files. The lower number of microcracks caused by Neoniti files compared with the other files, as evaluated in these studies, can be attributed to the novel manufacturing process, higher flexibility and microhardness of Neoniti files, as well as their square-shaped cross-section, which improves their cutting efficiency and applies a lower magnitude of load to root canal walls [37].

Kim et al. [38] also measured the amount of stress applied to the root canal walls by different rotary systems made of different alloys. Their finite element analysis showed that the design of self-adjusting NiTi files may produce minimal stress concentration on the root dentin of curved root canals compared to Profile and ProTaper files. In another study, Bier et al. [39] pointed to a direct association between the degree of rotation of a file and the number of cracks formed following its use. They concluded that use of single-file rotary systems compared with older systems applies a lower magnitude of load to root canal walls because the treatment process is accomplished by using a smaller number of files. Liu et al. [40] also found that single-file rotary systems cause smaller numbers of microcracks compared with using sequences of files, which can be due to the smaller number of procedures conducted on teeth with the use of single-file systems. This observation explains the absence of a significant difference in the number of microcracks in moderately curved and severely curved root canals after instrumentation with single-file systems in the present study. Harandi et al. [41] evaluated the number of microcracks created by ProTaper, Neoniti and SafeSider rotary systems and reported that the number of microcracks in the Neoniti group was higher than that in the other two groups. Furthermore, the maximum number of microcracks was noted in the coronal 9 mm of the canals. This finding may have been due to the method through which the microcracks were observed, since the authors used a stereomicroscope for this purpose; the method of preparation of teeth in this method increases the rate of errors. Furthermore, the difference in the size of the master apical file may also explain the difference in the results and the higher number of microcracks in the Neoniti group, since the authors used 0.25 Neoniti files while in the present study, 0.2 Neoniti files were used, according to the manufacturer’s instructions. The authors showed that all the rotary systems caused microcracks compared with the control group, which was in agreement with our findings. In contrast to previous studies, which mainly evaluated microcrack formation in single canal teeth [42,43,44], the present study was conducted on the mesiobuccal canals of mandibular molars, which are often narrow and feature buccolingual and mesiodistal curvature. This difference can also explain the significant increase in the number of microcracks after root canal instrumentation in the present study. On the other hand, Miguens-Vila et al. [30] also conducted research on mandibular first molars, but their results are contrary to those of our study. These differences could be due to their use of NiTi rotary files with single and multiple file systems, compared with our use of only the single-file system of Neolix, as well as the software they used to reconstruct their images, which featured a beam-hardening correction of 55%. Regarding the microcracks, Shemesh et al. [45] explained that microcracks are initiated during tooth extraction and can even be due to the desiccation of teeth. This statement supports our results, since some microcracks were seen on the micro-CT images of the teeth prior to the root canal instrumentation. This finding was also reported by De-Deus et al. [26,27], Ceyhanli et al. [9] and Bayram et al. [31]. Moreover, Ceyhanli et al. [9] reported that micro-CT provides images with higher resolution than stereomicroscopes, and factors such as lighting and the preparation technique used on teeth for assessment by stereomicroscope can increase the rate of errors. The anatomical assessment of the teeth in terms of geometry at baseline was a strength of this study, which increased the internal reliability of the results and eliminated anatomical biases that could have affected the results. Furthermore, the teeth in each of the two groups were standardized with regard to the degree of root canal curvature and the files were used with similar torques and speeds in all the teeth. Moreover, the micro-CT images were obtained under standardized conditions, which was another strength of this study. There is a need for further micro-CT studies to assess the effect of instrumentation on crack formation in curved root canals. The small sample size, the fact that other rotary systems were not assessed and manual filing was used, the fact that the numbers of microcracks in different sections along the roots were not compared, the fact that group pairing using the working length, surface area and volume was not performed and the lack of a discrete analysis, in which the number of microcracks are presented as a function of the severity of the root curvature, were among the limitations of this study. It is suggested that these approaches are considered in future studies.

## 5. Conclusions

Based on the results of this study, the instrumentation of moderately and severely curved root canals with Neoniti rotary files increases the number of microcracks; however, a higher degree of curvature does not necessarily translate to a higher number of microcracks.

## Figures and Tables

**Figure 1 materials-15-03002-f001:**
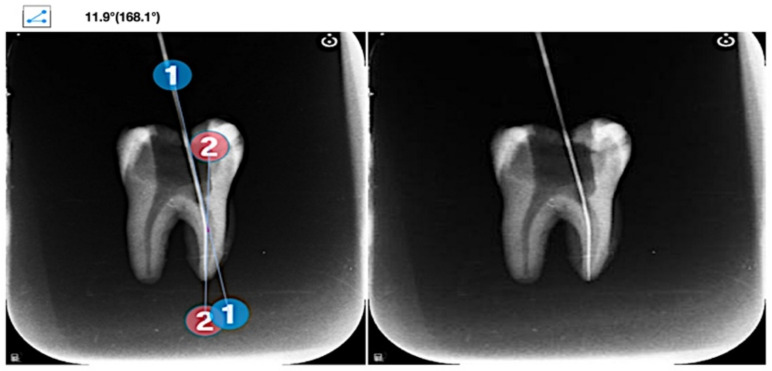
Measuring the root canal curvature on periapical radiographs according to Schneider’s method.

**Figure 2 materials-15-03002-f002:**
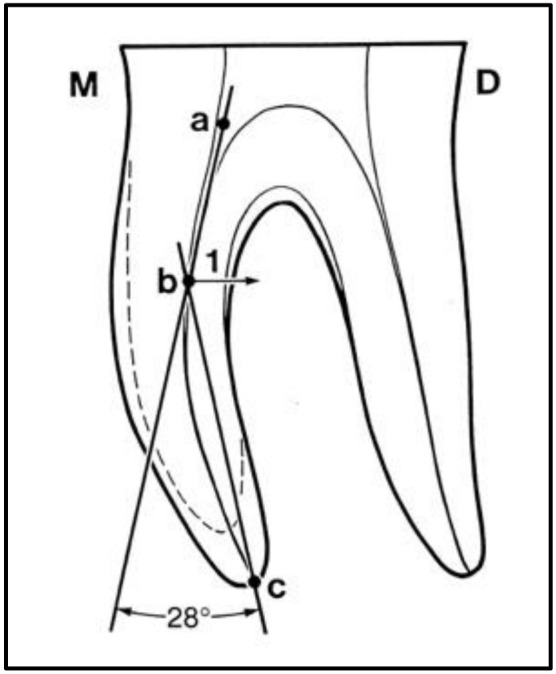
Schematic view of the root canal curvature measurement according to Schneider’s method. Point “a” was first identified on the file at the root canal orifice and a line was drawn from point “a” parallel to the file until the point of deviation (point “b”). Point “c” was then identified at the apical foramen, from which a line was drawn to point “b”. The angle formed between the two aforementioned lines indicates the degree of root canal curvature. M: mesial; D: distal; 1: A perpendicular line from the point b to the line ac gives the curvature height.

**Figure 3 materials-15-03002-f003:**
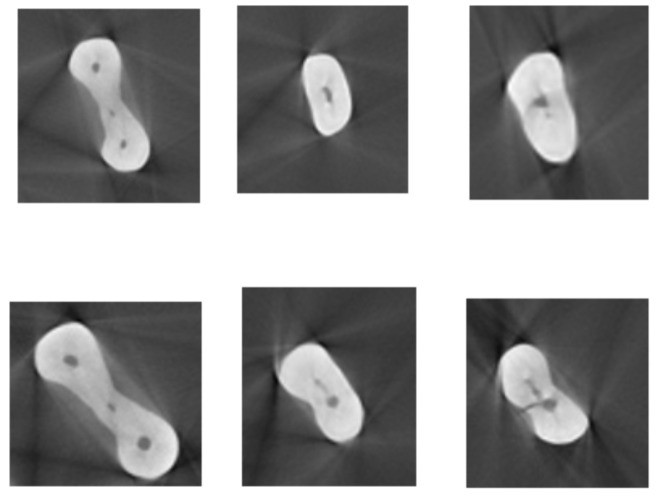
Sections of moderately curved mesial roots before (**up**) and after (**down**) instrumentation.

**Figure 4 materials-15-03002-f004:**
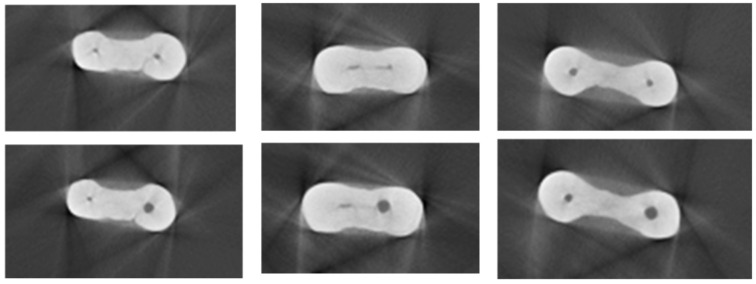
Three sections of severely curved mesial roots before (**up**) and after (**down**) instrumentation.

**Table 1 materials-15-03002-t001:** Measurements of central dispersion for the mean number of microcracks in the root canal walls of moderately curved canals before and after instrumentation (n = 9).

Time of Measurement	Minimum	Maximum	Mean	Std. Deviation
Preoperatively	0.00	18.52	7.40	8.02
Postoperatively	0.00	25.93	11.59	9.74

**Table 2 materials-15-03002-t002:** Measurements of central dispersion for the mean number of microcracks in the root canal walls of severely curved canals before and after instrumentation (n = 9).

Time of Measurement	Minimum	Maximum	Mean	Std. Deviation
Preoperatively	0.00	16.67	7.20	5.94
Postoperatively	6.67	23.33	13.23	5.64

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
