# Peer review of "Micro-Computed Tomographic Assessment of Microcrack Formation before and after Instrumentation of Curved Root Canals with Neoniti Rotary Files"

_materials, 2022, doi:10.3390/ma15093002_

Round 1

Reviewer 1 Report

Dear authors 

I think the explanations given helped to understand the remarks made. 
Thus from my point of view the manuscript should be accepted for publication.

Author Response

Dear Prof. Tabrizian,

Thank you very much for your response.

Please find kindly the answers to the queries below in *bold.

Dear authors 

I think the explanations given helped to understand the remarks made. 
Thus from my point of view the manuscript should be accepted for publication.

*Thank you.

Kind Regards

Reviewer 2 Report

Corrections are sufficient.

Author Response

Dear Prof. Tabrizian,

Thank you very much for your response.

Please find kindly the answers to the queries below in *bold.

Corrections are sufficient.

*Thank you.

Kind Regards

Reviewer 3 Report

I will limit my comments to one fundamental shortcoming of the manuscript: I cannot see any microcracks in the images.  The images are of poor quality, and the one shown in Figure 4 has severe motion artefacts indicating that the tooth was not held securely during the scan.  There is no mention of how the cracks were detected or any indication of their presence on the figures.  Without confidence in the quality of the data, any further analysis has no useful meaning.  Sorry to sound negative, the topic seems useful, although comparison perhaps with manual filing would add to its value. 

Author Response

Dear Prof. Tabrizian,

Thank you very much for your response.

Please find kindly the answers to the queries below in *bold.

I will limit my comments to one fundamental shortcoming of the manuscript: I cannot see any microcracks in the images.  The images are of poor quality, and the one shown in Figure 4 has severe motion artefacts indicating that the tooth was not held securely during the scan.  There is no mention of how the cracks were detected or any indication of their presence on the figures.  Without confidence in the quality of the data, any further analysis has no useful meaning.  Sorry to sound negative, the topic seems useful, although comparison perhaps with manual filing would add to its value. 

*The images has been changed.

*The cracks in the mesiobuccal root were detected under a ×10 magnification stereomicroscope equipped with a digital camera (SMP 200, HP, USA) from the mesial, buccal and lingual aspects.

*All micro-CT images were evaluated by Amira software, which is a powerful, multifaceted 2D-5D solution for visualization, processing and analysis of research data from many image modalities.  

*Comparison to manual filling is a good point and is added as a suggestion to the last paragraph of Discussion for further studies.

Kind Regards

Reviewer 4 Report

I congratulate the authors for conducting the present study regarding the micro crack formation on the process of root canal instrumentation.

Here goes a few of my concerns:

I suggest replacing the format “#20-6% Neoniti file” by “0.20/.06v Neoniti file”

In the paragraph “Third generation of the NiTi rotary files are manufactured by using heating and cooling procedures on wires and by applying M-wire and R-phase technologies and electrical discharge methods, which results in reduction of the separation risks of the file especially in curved root canals as well as high memory shape and reduction of cyclic fatigue and higher preservation of the initial anatomy of the root canal. Neoniti (Neolix, Châtres-la-Forêt, France) is one of the introduced instruments of this generation with full rotary motion and multiple taper in a single instrument that consist of one C1 for coronal enlargement and three A1 for shaping the canal (with #20, #25 and #40 tip sizes) files. [17-19]” I would recommend adding the manuscript DOI: 10.1016/j.joen.2020.07.016 since it makes a very good descripting of the characteristics of the A1 instrument.

How were the teeth selected?

How were the groups paired? Only by the curvature? The perfect groups pairing should be conducted using working length, surface area, volume and SMI.

How was this sample size decided?

How long were the teeth in dry environment and at which steps?

I recommend the authors to debate more the influence of drying periods of the crack formation. That has been shown as the main cause for the micro-cracks to appear.

I also recommend the authors to debate the limitation of the way the group pairing was conducted, which is not in accordance with previous studies.

Author Response

Dear Prof. Tabrizian,

Thank you very much for your response.

Please find kindly the answers to the queries below in *bold.

I congratulate the authors for conducting the present study regarding the micro crack formation on the process of root canal instrumentation.                                                        Here goes a few of my concerns:

I suggest replacing the format “#20-6% Neoniti file” by “0.20/.06v Neoniti file”                                       *The format has been corrected.

In the paragraph “Third generation of the NiTi rotary files are manufactured by using heating and cooling procedures on wires and by applying M-wire and R-phase technologies and electrical discharge methods, which results in reduction of the separation risks of the file especially in curved root canals as well as high memory shape and reduction of cyclic fatigue and higher preservation of the initial anatomy of the root canal. Neoniti (Neolix, Châtres-la-Forêt, France) is one of the introduced instruments of this generation with full rotary motion and multiple taper in a single instrument that consist of one C1 for coronal enlargement and three A1 for shaping the canal (with #20, #25 and #40 tip sizes) files. [17-19]” I would recommend adding the manuscript DOI: 10.1016/j.joen.2020.07.016 since it makes a very good descripting of the characteristics of the A1 instrument.                                                                              *The recommended manuscript has been added.

How were the teeth selected?                                                                                                            *Mandibular molars freshly extracted with two separate mesial canals and foramina due to several reasons were selected. Extracted mandibular molars with cracks, resorption or fracture, open apices, calcification of the mesiobuccal canal, internal root resorption and history of previous endodontic treatment were excluded from the study. After collection, the teeth were disinfected with 5.2% sodium hypochlorite and were inspected under a ×10 magnification stereomicroscope equipped with a digital camera (SMP 200, HP, USA) from the mesial, buccal and lingual aspects to ensure absence of cracks in the mesiobuccal root. Also radiographs of the teeth were made and the mesiobuccal canal curvature was measured after Schneider’s method.

How were the groups paired? Only by the curvature? The perfect groups pairing should be conducted using working length, surface area, volume and SMI.                                                                            *Pairing was performed regarding the root curvature. Teeth with 5-20° root canal curvature were assigned to the moderately curved group (n=9) while those with 20-40° root canal curvature were assigned to the severely curved group (n=9). *Pairing according to working length, surface area, volume and SMI is added in last paragraph of discussion as a suggestion for further studies.

How was this sample size decided?                                                                                                   *Statistical analysis indicated that n=9 should be the minimum sample size, considering α=0/05 and β=0/2 (power= 1-β= 80%) and considering differences up to 30% and more as standard deviation.

How long were the teeth in dry environment and at which steps?                                                                      *Upon collection, the teeth were disinfected with 5.2% sodium hypochlorite and were selected and finally stored in saline at 4°C until further examination to prevent dryness which may cause micro-cracks during storage. 

I recommend the authors to debate more the influence of drying periods of the crack formation. That has been shown as the main cause for the micro-cracks to appear.                                                                    *Thanks for your recommendation. This aspect is discussed.

I also recommend the authors to debate the limitation of the way the group pairing was conducted, which is not in accordance with previous studies.                                                                                        *Thanks. We also added this aspect to the discussion.

Kind Regards

Round 2

Reviewer 3 Report

In your repsponse, you say "*The cracks in the mesiobuccal root were detected under a ×10 magnification stereomicroscope equipped with a digital camera (SMP 200, HP, USA) from the mesial, buccal and lingual aspects."  However, in the text is says that the stereomicroscope was only used "to ensure absence of cracks in the mesiobuccal root" prior to instrumentation.

The methodology for counting cracks is not clear.  You say that "The number of images of each tooth showing microcracks was counted."  What are these images?  Is it one image per slice?  If so, the same crack might be seen in many slices.  Figure 3 relates to Figure 1 in Bayram et al [31].  However, in Bayram's figure, a crack is clearly seen and identified.  I would have far more confidence in this study if you could show a similar image with visible cracks identified.  My main problem with this is that I have no confidence that the measurement relates to anything real and not to an artefact.  I am sure that this is a very good study, but we need to see clearly how the results were derived and especially to see that the system is capable of resolving these cracks.  Can Figure 3 be swapped for one clearly showing cracks?  Perhaps the original figure does and these have been lost in the online image processing.  Bayram's micro-CT images are not great quality, but the cracks are still very visible.

Author Response

Dear Prof. Tabrizian,

Thank you very much for your response.

Please find kindly the answers to the queries below in *bold.

In your repsponse, you say "*The cracks in the mesiobuccal root were detected under a ×10 magnification stereomicroscope equipped with a digital camera (SMP 200, HP, USA) from the mesial, buccal and lingual aspects."  However, in the text is says that the stereomicroscope was only used "to ensure absence of cracks in the mesiobuccal root" prior to instrumentation.

​*The issue has been clarified in the text. The stereomicroscope has been used to 'ensure absence of cracks' and Micro CT was used for 'recognizing micro cracks'.

The methodology for counting cracks is not clear.  You say that "The number of images of each tooth showing microcracks was counted."  What are these images?  Is it one image per slice?  If so, the same crack might be seen in many slices.  Figure 3 relates to Figure 1 in Bayram et al [31].  However, in Bayram's figure, a crack is clearly seen and identified.  I would have far more confidence in this study if you could show a similar image with visible cracks identified.  My main problem with this is that I have no confidence that the measurement relates to anything real and not to an artefact.  I am sure that this is a very good study, but we need to see clearly how the results were derived and especially to see that the system is capable of resolving these cracks.  Can Figure 3 be swapped for one clearly showing cracks?  Perhaps the original figure does and these have been lost in the online image processing.  Bayram's micro-CT images are not great quality, but the cracks are still very visible.

​* The term images has been replaced by (images/(slices)). 'Figure 3' has been replaced by one clearly showing cracks.

​*The project was performed under the supervision of two professional Endodontist with about 30 years experience in the field of Endodontology, which can distinguish artifacts from real microcracks. This information has been added to the text.

Kind Regards

Reviewer 4 Report

Dear authors, I have no more concerns regarding the present manuscript. Thank you.

Author Response

Dear Prof. Tabrizian,

Thank you very much for your response.

Please find kindly the answers to the queries below in *bold.

Dear authors, I have no more concerns regarding the present manuscript. Thank you.

*Thank you.

Kind Regards